# *Tamarix hispida* NAC Transcription Factor *ThNAC4* Confers Salt and Drought Stress Tolerance to Transgenic *Tamarix* and *Arabidopsis*

**DOI:** 10.3390/plants11192647

**Published:** 2022-10-08

**Authors:** Meiheriguli Mijiti, Yucheng Wang, Liuqiang Wang, Xugela Habuding

**Affiliations:** 1Xinjiang Key Laboratory of Special Species Conservation and Regulatory Biology, Key Laboratory of Special Environment Biodiversity Application and Regulation in Xinjiang, Key Laboratory of Plant Stress Biology in Arid Land, College of Life Science, Xinjiang Normal University, Urumqi 830054, China; 2State Key Laboratory of Tree Genetics and Breeding, Northeast Forestry University, Harbin 150040, China; 3State Key Laboratory of Tree Genetics and Breeding, Research Institute of Forestry, Chinese Academy of Forestry, Beijing 100091, China

**Keywords:** abiotic stress, *Tamarix hispida*, ROS scavenging, NAC transcription factor

## Abstract

Salt and drought are considered two major abiotic stresses that have a significant impact on plants. Plant NAC (NAM, ATAF1/2, and CUC2) transcription factors (TFs) have been shown to play vital roles in plant development and responses to various abiotic stresses. *ThNAC4*, a NAC gene from *Tamarix hispida* involved in salt and osmotic stress tolerance, was identified and characterized in this study. According to a phylogenetic study, *ThNAC4* is a member of NAC subfamily II. Subcellular localization analysis showed that *ThNAC4* is located in the nucleus, and transcriptional activation experiments demonstrated that *ThNAC4* is a transcriptional activator. Transgenic *Arabidopsis* plants overexpressing *ThNAC4* exhibited improved salt and osmotic tolerance, as demonstrated by improved physiological traits. *ThNAC4*-overexpressing and *ThNAC4*-silenced *T. hispida* plants were generated using the transient transformation method and selected for gain- and loss-of-function analysis. The results showed that overexpression of *ThNAC4* in transgenic *Tamarix* and *Arabidopsis* plants increased the activities of antioxidant enzymes (SOD, POD, and GST) and osmoprotectant (proline and trehalose) contents under stress conditions. These findings suggest that *ThNAC4* plays an important physiological role in plant abiotic stress tolerance by increasing ROS scavenging ability and improving osmotic potential.

## 1. Introduction

Plants grow in dynamic environments in which factors such as drought, salt, extreme temperature, and biotic stresses frequently constrain growth and development. Among these stresses, salt and drought are two major stresses that have a significant impact on plants. To combat and survive these stresses, plants have developed several adaptive mechanisms involving signal transduction and gene expression regulation.

Transcription factors (TFs) are DNA-binding proteins that regulate the expression of downstream genes as *trans*-acting elements via specific binding to the *cis*-acting elements present in the promoter region of the target genes [1]. Many transcription factors, such as AP2/ERF, bZIP, MYB, WRKY, and NAC, have been found to play critical roles in the plant response to various abiotic stimuli [2,3,4,5,6]. Among them, NAC (NAM, ATAF1/2, and CUC2) TFs are one of the largest plant-specific TF families that play important roles in the regulation of plant abiotic stress responses. For instance, in *Arabidopsis*, *ANAC019, ANAC055*, and *ANAC072* are induced by drought, salinity, and abscisic acid (ABA), and overexpression of these NAC genes confers enhanced drought tolerance in transgenic *Arabidopsis* [7,8]. *OsNAC5*, *OsNAC6*, and *OsNAC10*, three ABA-dependent NAC TFs from rice, improve drought tolerance in transgenic plants by regulating stress-related genes [9,10]. *TaNAC2, TaNAC67*, or *TaNAC69* gene overexpression in *Arabidopsis* has been shown to improve tolerance to salt and drought stresses [11,12,13]. Transgenic *Arabidopsis* overexpressing *MbNAC29* from *Malus baccata* was shown to improve cold and high salinity tolerance by increasing the scavenging capability of reactive oxygen species (ROS) [14]. A cotton NAC transcription factor *GhirNAC2* was found to promote tolerance to drought stress by regulating ABA biosynthesis and stomatal closure [15]. The overexpression of *GmNAC06* induced tolerance to salinity in soybeans [16]. *CaNAC46* in *Capsicum annuum* has been identified as a positive regulator that activates ROS-scavenging enzymes and enhances root formation under salt and drought conditions [17]. Recently, 12 *HaNAC* genes were identified by genome-wide identification and found to be involved in the response to salt and drought stress in sunflowers [18]. Although NAC proteins have been studied in various plant species for their roles in abiotic stress, little is known about the biological functions of NAC TFs in abiotic stress tolerance in halophytic woody plants. Furthermore, it has been proposed that the exploitation of halophytes could be an excellent strategy for understanding the physiological/molecular mechanisms of abiotic stress tolerance, leading to the identification of novel candidate genes for breeding stress-tolerant plants [19]. Therefore, understanding the molecular mechanisms of halophytes in response to abiotic stress should provide a scientific foundation for developing stress-resistant varieties.

*Tamarix hispida*, a species of woody halophyte that is highly tolerant to salinity, drought, and extreme temperatures, is widespread in the saline soils of drought-stricken areas of Central Asia and China [20]. Due to these characteristics, the species is an excellent candidate for research into stress tolerance mechanisms. In a previous study, 21 full-length coding regions of *T. hispida* NAC genes were discovered, and *ThNAC4* was primarily upregulated in response to salt and drought stresses [21]. In the present study, we cloned the *ThNAC4* gene from *T. hispida* and characterized its function in salt and drought stress tolerance. The results showed that *ThNAC4* positively regulates salt and osmotic tolerance by increasing ROS-scavenging ability and enhancing osmolyte accumulation. The findings enhanced our knowledge of NAC TFs in the abiotic stress response, which helped to lay the groundwork for breeding stress-tolerant plants with genetic engineering.

## 2. Results

### 2.1. Cloning and Bioinformatics Analysis of ThNAC4 from Tamarix hispida

The open reading frame (ORF) of *ThNAC4* is 1158 bp in length and is predicted to encode a protein of 387 amino acid residues with a molecular weight of 42.65 kDa. Multiple sequence alignment analysis revealed that ThNAC4 and NAC proteins from other plant species shared a highly conserved binding domain at the N-terminus, which is the basic characteristic of the NAC family (Appendix A). A phylogenetic tree was constructed to study the evolutionary relationships between ThNAC4 and 94 other NACs from *Arabidopsis*. This analysis showed that ThNAC4 is closely related to two *Arabidopsis* NAC proteins, ANAC080/ATNAC4 (AT5G07680) and ANAC100/ATNAC5 (AT5G61430), from NAC subfamily II (Figure 1), suggesting that ThNAC4 should also belong to NAC subfamily II.

### 2.2. ThNAC4 Is Localized in the Nucleus and Exhibits Transactivation Activity

The N-terminus of NAC family proteins has a highly conserved domain that might serve as a nuclear localization signal (NLS). Numerous NACs are located in the nucleus of cells [22,23]. To determine whether *ThNAC4* is located in the nucleus, we transformed ThNAC4-GFP fusion proteins into onion epidermal cells by the *Agrobacterium*-mediated transformation method, and 35S-GFP functioned as a control. As shown in Figure 2A, free GFP was distributed uniformly in the cytoplasm, whereas the ThNAC4-GFP fusion proteins were detected in the nuclei. This finding indicates that ThNAC4 is a nuclear protein and supports the idea that ThNAC4 functions as a transcription factor.

Transcriptional activation ability was investigated according to Hu et al. [24]. The entire CDS or serial deletions of the *ThNAC4* CDS were combined with pGBKT7 plasmids and transformed into Y2HGold cells to detect their transactivation activity (Figure 2B). All transformants could grow normally on the SD/-Trp plates, showing that they had been transformed into Y2HGold cells. The yeast cells containing the full-length CDS of *ThNAC4* grew well and appeared blue on the SD/-Trp/-His/X-a-Gal plates, suggesting that ThNAC4 functions as a transcriptional activator. Furthermore, we noticed that the yeast cells transformed with the predicted C-terminal activation domain (amino acids 182–386) of ThNAC4 thrived and became blue, whereas cells expressing the N-terminal domain (amino acids 1–193) of ThNAC4 were not blue, showing that the C-terminal domain of ThNAC4 has transactivation activity. In addition, analysis of the truncated CDSs of ThNAC4 suggested that ThNAC4 contains the transcriptional activation domain at the region of amino acids 175–291.

### 2.3. Constitutive Expression of ThNAC4 Enhances Salt and Osmotic Resistance in Transgenic Arabidopsis

To reveal the molecular function of *ThNAC4* in vivo under salt and osmotic stress, we obtained seven independent T3 homozygous *Arabidopsis* lines overexpressing *ThNAC4*. Three independent transgenic *Arabidopsis* lines (OE2, OE8, and OE9) with high transcript levels of *ThNAC4* were selected for further experiments (Appendix A). Under normal growth conditions, the seed germination, growth phenotype, fresh weight, and root length were not different between transgenic *Arabidopsis* and wild-type (WT) plants (Figure 3). However, after treatment with NaCl (150 mmol/L) and mannitol (200 mmol/L), the three tested transgenic lines were greener and exhibited less wilting than WT plants, and they displayed significantly improved seed germination rates, root length, and fresh weight (Figure 3B–F). In addition, the chlorophyll contents were similar among the three transgenic lines and WT plants under normal conditions. However, under salt and osmotic stress conditions, all transgenic lines had significantly higher chlorophyll levels than WT plants (Figure 3G). Moreover, the water loss assay indicated that the water loss rates of transgenic *Arabidopsis* were lower than those of WT plants (Figure 3H). These results together indicated that the constitutive expression of *ThNAC4* conferred enhanced tolerance to NaCl and mannitol treatment in transgenic plants.

Abiotic stresses can lead to oxidative damage due to the increased production of reactive oxygen species (ROS). Therefore, plants need to activate antioxidative systems to cope with oxidative damage. To determine the accumulation of two main ROS species, superoxide anion (O_2_^-^) and hydrogen peroxide (H_2_O_2_), in transgenic and WT *Arabidopsis* plants under osmotic and salt stresses, nitroblue tetrazolium (NBT) and diaminobenzidine (DAB) staining were performed. There were no significant differences in O_2_^-^ and H_2_O_2_ levels between transgenic lines and WT plants under normal growth conditions. However, both the O_2_^-^ and H_2_O_2_ accumulation levels were lower in transgenic lines than in WT plants under both osmotic and salt stress conditions (Figure 4A,B). We determined the level of cell death in leaves under salt and osmotic stress conditions using Evans blue staining. The results showed that the level of cell membrane damage was lower in the transgenic lines than in the WT plants under both salt and osmotic stress conditions (Figure 4C). The malondialdehyde (MDA) content was analyzed to investigate membrane lipid peroxidation levels. Likewise, the transgenic *Arabidopsis* accumulated less MDA than WT under salt and osmotic stresses (Figure 4D). In addition, the activities of the two main antioxidant enzymes, superoxide dismutase (SOD) and peroxidase (POD), induced by NaCl and mannitol treatments were analyzed in transgenic and WT plants. The activities of SOD and POD were significantly higher in transgenic lines than in WT plants under both salt and osmotic stresses (Figure 5E,F). Glutathione S-transferases (GSTs) are involved in catalyzing detoxification processes in plant development and plant stress responses, including GSH-dependent POD reactions that scavenge toxic organic hydroperoxides [25]. In this study, we also measured GST activity in transgenic and WT *Arabidopsis* plants. GST activity in all transgenic plants was higher than that in WT plants under salt and osmotic stress conditions (Figure 4G), indicating that overexpression of *ThNAC4* enhanced GST activity in transgenic *Arabidopsis*. These results indicated that overexpression of *ThNAC4* could improve ROS scavenging enzyme activity to lower ROS accumulation and limit cell damage in transgenic *Arabidopsis* under salt and osmotic stress conditions.

We next studied whether *ThNAC4* is involved in osmotic protectant and compatible solute accumulation under salt and osmotic stress conditions. The proline and trehalose contents were measured in comparison to the WT and transgenic *Arabidopsis*. The results showed that there was no distinct difference in proline and trehalose contents between transgenic and WT *Arabidopsis* plants under normal growth conditions. However, under conditions of salt and osmotic stresses, the contents of these osmolytes were significantly higher in transgenic *Arabidopsis* plants than in WT plants (Figure 4H,I), suggesting that *ThNAC4* plays a role in the regulation of osmotic protectant and compatible solute biosynthesis.

Additionally, we studied the genes involved in SOD, POD, and trehalose biosynthesis in transgenic and WT *Arabidopsis* plants with NaCl and mannitol treatments using the qRT-PCR method. The results showed that the expression of many studied genes was highly increased in *ThNAC4*-transformed *Arabidopsis* under NaCl and mannitol treatments (Figure 5). These results indicate that *ThNAC4* can increase the transcription of the *SOD, POD*, *TPS*, and *TPP* genes, which may contribute to increased SOD and POD activities and trehalose content under salt and drought stress.

### 2.4. Transient Overexpression or Knockdown of ThNAC4 in T. hispida Plants

To further study the function of *ThNAC4* using gain- and loss-of-function methods, we generated *ThNAC4*-overexpressing (OE), RNAi-silenced *ThNAC4* (IE), and control plants (transformed with empty pROKII, VC) and examined the expression of *ThNAC4* using the qRT-PCR method. The expression of *ThNAC4* was significantly increased in the OE plants but was significantly decreased in the IE plants compared to those in the control (VC) plants (Appendix A). These results indicated that *ThNAC4* had been successfully overexpressed or knocked down and was suitable for further study.

### 2.5. ThNAC4 Affects ROS Scavenging Capability

ROS accumulation is a substantial indicator of the degree of stress tolerance. Therefore, NBT and DAB staining were used to determine two major ROS species, superoxide anion (O_2_^-^) and hydrogen peroxide (H_2_O_2_), in these transgenic *T. hispida* plants. Under salt and osmotic stress conditions, compared with the control plants, OE *T. hispida* plants exhibited reduced NBT and DAB staining, while IE *T. hispida* plants displayed significantly increased NBT and DAB staining (Figure 6A). Moreover, the MDA contents were measured in transgenic *T. hispida* plants. Under normal growth conditions, OE *T. hispida* plants had lower MDA levels than control and IE *T. hispida* plants. When exposed to salt or osmotic stress, IE *T. hispida* plants had significantly higher MDA levels, whereas OE plants had significantly lower MDA levels than control plants (Figure 6B). In addition, OE *T. hispida* plants also displayed higher chlorophyll contents than control and IE plants (Figure 6C).

We further studied the activities of three primary ROS scavenging enzymes, SOD, POD, and GST, in the three types of transiently transformed *T. hispida* plants. Under salt and osmotic stress conditions, the OE plants had significantly higher SOD, POD, and GST activities than the control and IE plants (Figure 6D,E). These results indicate that *ThNAC4* expression enhances ROS scavenging capability by increasing the activities of antioxidant enzymes.

### 2.6. Analysis of Proline and Trehalose Contents

The proline and trehalose contents were measured in all transgenic *T. hispida* plants to study whether *ThNAC4* is involved in the biosynthesis of these osmolytes in salt or osmotic stress environments. As shown in Figure 6G,H, there was no significant difference in proline and trehalose contents between the three types of transgenic *T. hispida* seedlings under normal conditions. However, under salt and osmotic stress conditions, the OE *T. hispida* plants had significantly higher proline and trehalose levels, followed by the IE and control plants. These results are consistent with the results from transgenic *Arabidopsis*, indicating that overexpression of *ThNAC4* increases osmolyte contents under salt and osmotic stresses.

## 3. Discussion

Salt and drought are two major stresses that have a significant impact on plants. When exposed to drought stress, plants display a reduced photosynthetic rate and lower chlorophyll content. Furthermore, long-term drought stress also leads to oxidative damage because of increased ROS accumulation, which disrupts cellular physiological homeostasis [26]. The first stage of salinity stress restricts plant growth in the form of osmotic stress, which is then followed by ion toxicity, which causes various physiological changes in plant cells [27].

Plants have developed various defense mechanisms to combat the negative impacts of abiotic stress, and transcriptional control of gene expression plays an important role in this process. Plant-specific NAC TFs are one of the key regulators in plant responses to various abiotic stresses, and their function has been studied in a variety of plant species [28]. However, few studies have investigated the abiotic tolerance function of NAC TFs in halophytic woody plants. Therefore, identifying and characterizing novel stress-responsive NAC TFs from halophytes such as *T. hispida* can provide more insight into this special group of transcription factors.

Previously, a total of 21 full-length NAC genes were identified in *T. hispida*. qRT-PCR analysis revealed that these *ThNAC* genes are all expressed in the roots, stems, and leaves of *T. hispida* in response to salinity, drought, heavy metal, or ABA stimuli, suggesting that they play important roles in the abiotic stress response and are involved in ABA-dependent stress-signaling pathways [21]. Therefore, we selected *ThNAC4* for further research. Multiple sequence alignment analysis indicated that *ThNAC4* contains a classic NAC domain that is the basic characteristic of the NAC family. Based on phylogenetic tree analysis, *ThNAC4* falls into NAC subfamily II (Figure 1) and is most similar to *ATNAC4 (AT5G07680)* gene in *Arabidopsis*, which responds to biotic and abiotic stresses [29,30]. According to previous studies, *ThNAC4* was included in NAC subfamily I in *T. hispida*, which contains *ThNAC13* and *ThNAC7*. Moreover, these NAC genes were found to enhance tolerance to salt and osmotic stress by improving the osmotic potential and increasing ROS scavenging [31,32]. Therefore, we decided to further study the role of *ThNAC4* in salt and osmotic stresses.

Transcriptional activation assays were performed to investigate whether ThNAC4 activated transcription and to identify the activation domain. The results suggested that ThNAC4 is a transcriptional activator and that the transcriptional activation domain is located in a region containing amino acids 175–291 of ThNAC4 (Figure 2).

Previous research has shown that NAC TFs play important roles in the response to both salinity and drought and identified increased expression of genes associated with tolerance to these stresses [28,32,33]. Therefore, three independent *ThNAC4* transgenic *Arabidopsis* lines (OE2, OE5, and OE9) and three types of transiently transformed *T. hispida* (OE, IE, and VC) were used to analyze the molecular function of *ThNAC4* under salt and osmotic stresses. Growth phenotype, seed germination, root length, fresh weight, and chlorophyll content are important markers for evaluating stress tolerance in plants. Our results showed that overexpression of *ThNAC4* significantly improved these physiological parameters in both transgenic *Arabidopsis* and *T. hispida* plants (Figure 3 and Figure 6), suggesting that *ThNAC4* plays a positive role in plant tolerance to salt and osmotic stress.

Reactive oxygen species (ROS) are involved in the oxidation of requisite biomolecules and act as a signal under abiotic stress conditions [34]. However, plants accumulate excessive ROS when exposed to an adverse environment, which causes imbalances in homeostasis at the cellular level and, eventually, cell death. Therefore, antioxidant defense systems play an important role in plant abiotic stress resistance [35]. According to a previous study, NAC TFs generally share a positive relationship with the activation of plant antioxidant systems [14,17]. We found that overexpression of *ThNAC4* could reduce excess ROS accumulation under salt and osmotic stress conditions (Figure 4 and Figure 6). Moreover, the transcription levels of *SOD* and *POD* genes were induced by *ThNAC4* in transgenic *Arabidopsis*, which corresponded with the activities of SOD and POD, respectively. Plant GSTs are involved in protecting plants against different abiotic stress conditions [25]. A previous study overexpressing *SINAC2* in *Arabidopsis* showed enhanced salt and drought tolerance with alterations in glutathione metabolism [36]. In this study, overexpression of *ThNAC4* in *Arabidopsis* and *T. hispida* increased the content of GST under both salt and osmotic stress conditions, suggesting that it may also be involved in the detoxification of excess ROS in plants. These results suggested that *ThNAC4* increases stress tolerance by at least partially reducing ROS accumulation and membrane lipid peroxidation by enhancing the activities of antioxidant enzymes and ROS-scavenging capability.

The resilience of woody halophytes mainly relies on osmoregulation. The production and dislocation of osmolytes as part of the osmoregulatory process is critical for moderate stress tolerance [37]. Under stress conditions, many plants accumulate proline as a nontoxic and protective osmolyte [38]. It contributes to the stabilization of subcellular structures, buffering cellular redox potential, maintaining photosynthetic activity, scavenging free radicals, modulating cellular functions, and even triggering gene expression [39]. Trehalose is a typical organic osmolyte that is effectively involved in plant abiotic stress tolerance. It is a natural nonreducing sugar that plays a key role as a carbon source, an osmoprotectant, and a stabilizing molecule in plants [40]. Trehalose is synthesized in plants in a two-step process. Trehalose phosphate synthase (TPS) catalyzes the formation of T6P from UDP-glucose and glucose-6-phosphate. T6P is dephosphorylated by trehalose phosphate phosphatase (TPP) to generate trehalose. In *Arabidopsis thaliana*, there are 11 genes encoding TPSs (*AtTPS1-11*) and 10 genes encoding TPPs (*AtTPPA-J*) [41]. Our results showed that overexpression of *ThNAC4* significantly increased the proline and trehalose contents in transgenic *Arabidopsis* plants under salt and osmotic stress conditions (Figure 4). At the same time, transcripts of *ThNAC4* were positively correlated with the proline and trehalose contents (Figure 6). In addition, we further investigated the expression levels of *TPS* and *TPP* genes related to trehalose biosynthesis in *Arabidopsis*. *ThNAC4* overexpression in *Arabidopsis* plants increased the transcription levels of many studied genes under NaCl and mannitol treatments, which may play important roles in the biosynthesis of trehalose when the plant is exposed to salt and osmotic stresses (Figure 4 and Figure 5). These findings suggested that *ThNAC4* may activate proline and trehalose biosynthesis to increase the contents of these osmolytes, thereby enhancing plant abiotic stress tolerance by increasing osmotic potential and ROS-scavenging capability.

## 4. Materials and Methods

### 4.1. Plant Materials and Growth Conditions

*Tamarix hispida* seeds were cultured in tissue culture bottles with half-strength Murashige-Skoog (1/2 MS) solid medium [2% (w/v)] in a greenhouse at 24 °C, 70–75% relative humidity, and a 14 h light/10 h dark photoperiod. *Arabidopsis thaliana* Columbia (Col-0) seeds were germinated on 1/2 MS solid medium plates. After stratification in the dark at 4 °C for two days, the plates were placed in a greenhouse for germination. One-week-old seedlings were transferred from the plates to pots containing a 4:1:1 mixture of soil/vermiculite/perlite and grown in the greenhouse at 22 °C, 70–75% relative humidity, and a 16 h light/8 h dark photoperiod.

### 4.2. Cloning and Sequence Analysis of ThNAC4

The cDNA sequence of *ThNAC4* (GenBank number: JQ974958) was identified from the transcriptomes of *T. hispida* [21]. CLUSTALX1.81 was used to perform multiple sequence alignments of ThNAC4 and 12 NAC proteins from different species. The phylogenetic tree of ThNAC4 and 94 NAC proteins from *Arabidopsis* was generated with MEGA7 software.

### 4.3. Subcellular Localization Analysis of the ThNAC4 Protein

The full-length coding region of *ThNAC4* without the termination codon was fused with the N-terminus of green fluorescent protein (GFP) in the pBI121 vector driven by the CaMV 35S promoter to generate the 35S-ThNAC4-GFP construct. The GFP gene driven by a 35S promoter (35S-GFP) was used as a control. The two kinds of plasmids were transformed into onion epidermal cells by the *Agrobacterium*-mediated transformation method [42]. The onion epidermal cells were treated on MS medium plates in the dark at 25 °C for 48 h and visualized using confocal laser scanning microscopy (LSM800, Zeiss, Jena, Germany).

### 4.4. Transactivation Activity Analysis in Yeast

Transactivation activity analysis was performed according to Hu et al. [24]. To investigate the transactivation activity of *ThNAC4*, the entire coding region and a series of truncated *ThNAC4* genes were cloned into the pGBKT7 vector (Clontech, Palo Alto, CA, USA) and fused with a GAL4 DNA-binding domain. The yeast strain Y2H (Clontech, Palo Alto, CA, USA) was transformed with eight constructs and the pGBKT7 vector (negative control). The transformed strains were confirmed by sequencing and streaked on SD/-Trp/-His/X-α-gal plates. The trans-activation activities were evaluated based on their growth status. The plates were incubated for 3–5 days at 30 °C. Three independent experiments were performed.

### 4.5. Plasmid Construction and Plant Transformation

To obtain *ThNAC4*-overexpressing plants, the open reading frame (ORF) of *ThNAC4* was inserted into pROKII and stably transformed into *Arabidopsis thaliana* by using the floral dip method [43]. The highly efficient transient transformation of *T. hispida* plants was performed according to Ji et al. [44]. We constructed three types of transgenic *T. hispida* plants: *ThNAC4* overexpression plants (OE, transformed with 35S: ThNAC4), *ThNAC4* inhibited expression plants (IE, transformed with pFGC: ThNAC4), and control *T. hispida* plants (VC) with the empty pROKII plasmid.

### 4.6. Stress Tolerance Assays

Three T3 generation lines (Line OE2, OE8, and OE9) of *ThNAC4*-transformed *Arabidopsis* plants were employed in the salt and drought stress tolerance assays. For the germination rate assay, the seeds of wild-type (WT) and transgenic *Arabidopsis* plants were surface sterilized and sown on 1/2 MS medium (as a control) and 1/2 MS medium containing 150 mM NaCl or 200 mM mannitol. The plates were chilled at 4 °C in the dark for 2 days for stratification and were transferred to the greenhouse for 7 days. The germination rates of the transgenic and WT lines were calculated. For the stress tolerance experiment, seeds were sown on 1/2 MS medium for 7 days for germination and then transferred into 1/2 MS medium or 1/2 MS medium supplied with 150 mM NaCl or 200 mM mannitol. After seven days, the root length and fresh weight were measured, and photographs of the seedlings were taken. *Arabidopsis* seeds were planted in the soil after five days of culture in 1/2 MS plates. Three-week-old seedlings were watered with 150 mM NaCl or 200 mM mannitol for one week, and the chlorophyll contents and water loss of the plants were determined. The chlorophyll contents of detached leaves were measured as described by Lichtenthaler [45]. Three independent experiments were performed.

### 4.7. Physiological Analysis of the Stress Response 

For histochemical staining analysis, the transformed *T. hispida* seedlings were treated with 150 mM NaCl or 200 mM mannitol for 24 h. Four-week-old *Arabidopsis* plants grown in soil were watered with 150 mM NaCl or 200 mM mannitol solution for 2 h. The leaves of *T. hispida* or *Arabidopsis* plants were stained with nitroblue tetrazolium (NBT) and diaminobenzidine (DAB) using the methods of Zhang et al. [46] and Fryer et al. [47], respectively. Evans blue staining for examination of cell death was performed as described by Kumar et al. [48]. For physiological assays, *T. hispida* seedlings were cultured for 48 h in 1/2 MS or 1/2 MS with 150 mM NaCl or 200 mM mannitol, and *Arabidopsis* grown in the soil for four weeks was watered with 150 mM NaCl or 200 mM mannitol solution for 48 h. The malondialdehyde (MDA) content was determined using an MDA content measuring kit (Nanjing Jiancheng Bioengineering Institute, Nanjing, China). The superoxide dismutase (SOD), peroxidase (POD), and glutathione S-transferase (GST) activities were determined using SOD, POD, and GST enzyme activity measuring kits, respectively (Nanjing Jiancheng Bioengineering Institute, Nanjing, China). Proline was measured using a proline content measuring kit (Nanjing Jiancheng Bioengineering Institute, Nanjing, China). Trehalose content was measured using a trehalose content measuring kit (Suzhou Keming Bioengineering Institute, Suzhou, China). Three independent experiments were performed.

### 4.8. Real-Time Quantitative RT-PCR Assay

*T. hispida* total RNA was isolated using a CTAB (hexadecyltrimethylammonium bromide) method with slight adjustments described by Chang et al. [49] *Arabidopsis* total RNA was isolated using the Plant Total RNA Extraction Kit. The Primescript RT Reagent Kit was used to reverse-transcribe total RNA (1 μg) into cDNA using oligo (dT) primers, which were then diluted to 100 μL as a PCR template. The MJ Research OptionTM2 instrument (Bio-Rad, Hercules, CA, USA) was used for real-time PCR. To normalize the number of templates used in PCR for *T. hispida* analysis, *α-tubulin* (GenBank number: FJ618518), *β-tubulin* (GenBank number: FJ618519), and *β-actin* (GenBank number: FJ618517) were used as the internal references. *Tub2* (GenBank number: AT5G62690) and *Act7* (GenBank number: AT5G09810) were used as the internal references for *Arabidopsis* analysis. RT-PCR was performed using GoTaq qPCR Master Mix and the Step One Real-Time PCR System (TransGen Biotech, Beijing, China). The reaction system contained 2 μL of cDNA template, 0.5 μM of each primer, and 10 μL of SYBR Green Real-time PCR Master Mix in a 20 μL volume. The amplification was conducted at the following temperatures: 94 °C for 30 s, 45 cycles of 94 °C for 12 s, 58 °C for 30 s, 72 °C for 40 s, and 80 °C for 1 s for plate reading. Three biological replicates were conducted, and the expression levels were calculated using the 2^−ΔΔCT^ method [50]. Three independent experiments were performed. Appendix A lists all the primer sequences used in this study.

### 4.9. Statistical Analysis

Data analyses were conducted using SPSS statistical software (Version 19). Significance was calculated using Student’s *t* test, with a threshold of *p* < 0.05. * represents *p* < 0.05 in the figures. Figures were created using GraphPad Prism 5.

## 5. Conclusions

NAC TFs play an important role in plant responses to abiotic stress. In this study, a stress-responsive NAC gene from *T. hispida*, *ThNAC4*, was cloned and functionally characterized. According to our results, *ThNAC4* functions as a transcription factor that positively regulates salt and osmotic tolerance by reducing ROS accumulation and membrane damage by increasing antioxidant enzyme activities, ROS-scavenging ability, and osmolyte accumulation. Although this study contributed to fundamental understanding of the role of *ThNAC4* in the *T. hispida* response to salt and drought stresses, further research is still required to fully understand the detailed molecular mechanism of *ThNAC4* in the complex antioxidative system and metabolic pathways under abiotic stress.

## Figures and Tables

**Figure 1 plants-11-02647-f001:**
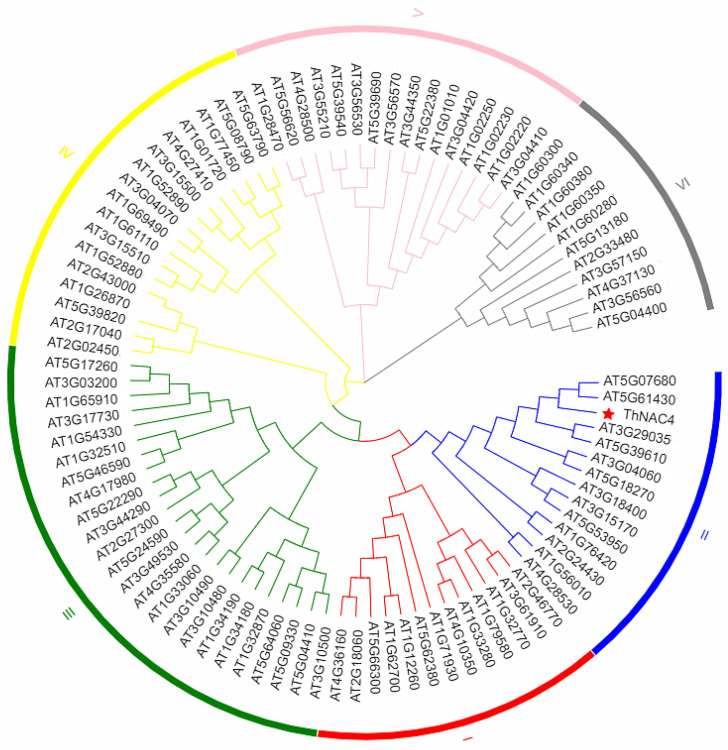
Phylogenetic analysis of ThNAC4 and NAC proteins from *Arabidopsis*. The phylogenetic relationship of *ThNAC4* and NAC proteins from *Arabidopsis*. *ThNAC4* and the *Arabidopsis* NACs were aligned; the unrooted NJ tree was constructed using MEGA 7.

**Figure 2 plants-11-02647-f002:**
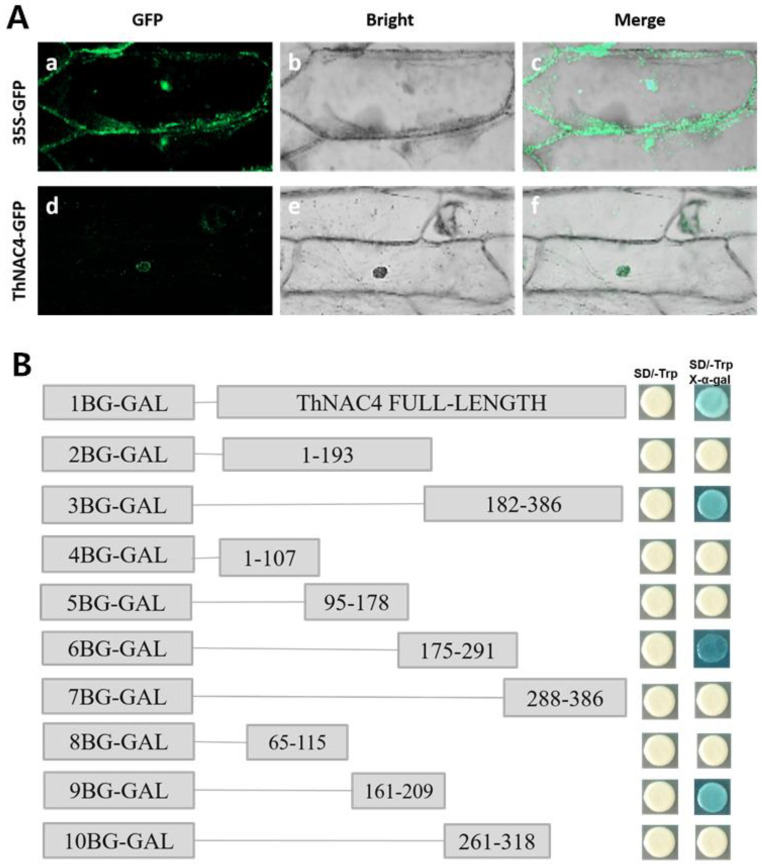
Subcellular localization and transcriptional activation of ThNAC4. (**A**) Subcellular localization analysis of ThNAC4. The 35S-GFP (control) and ThNAC4-GFP fusion proteins were transiently expressed in onion epidermal cells and viewed with a confocal microscope (wavelength 488 nm). (**a**) 35S-GFP, GFP; (**b**) 35S-GFP, bright; (**c**) 35S-GFP, merge; (**d**) ThNAC4-GFP, GFP; (**e**) ThNAC4-GFP, bright; (**f**) ThNAC4-GFP, merge. (**B**) Transactivation assay of ThNAC4. Full-length or truncated CDSs of ThNAC4 were cloned into the pGBKT7 vector, transformed into Y2HGold cells and grown on SD/-Trp or SD/-Trp/-His/X-α-gal media to assess their transcriptional activation.

**Figure 3 plants-11-02647-f003:**
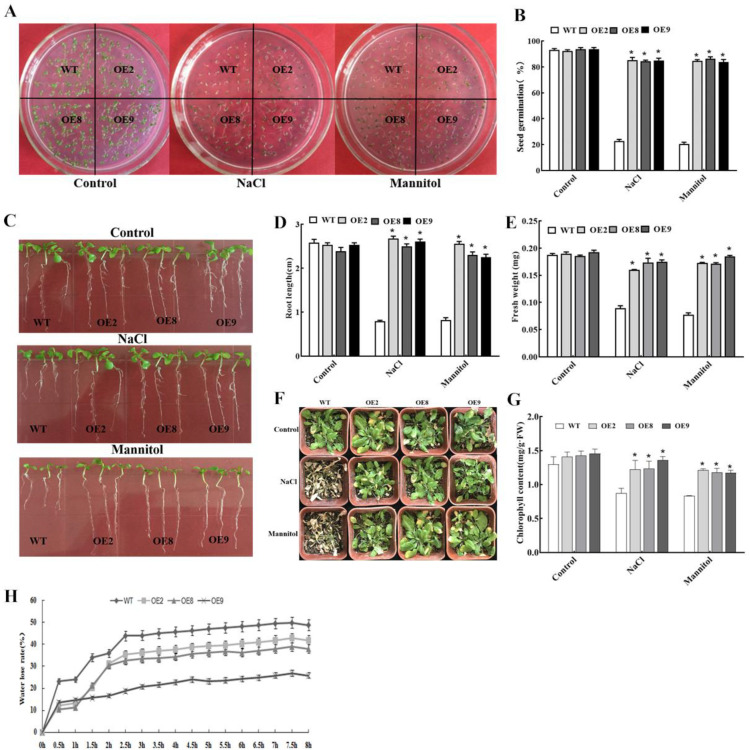
Stress tolerance of *Arabidopsis* overexpressing *ThNAC4*. (**A**,**B**) Seed germination assay of transgenic (OE2, OE8 and OE9) and wild-type (WT) *Arabidopsis* plants. Seeds were sown on 1/2 MS solid medium containing 150 mM NaCl or 200 mM mannitol and incubated at 22 °C for 7 days. Photographs were taken, and the germination rates were measured. (**C**–**E**) Effects of salt and drought stresses on root length and fresh weight. Seven-day-old seedlings were grown on 1/2 MS solid medium containing 150 mM NaCl or 200 mM mannitol for 7 days. Photographs were taken, and the root length and fresh weight were measured. (**F**) Growth phenotype of transgenic and WT *Arabidopsis* plants. (**G**,**H**) Effects of salt and drought stress treatment on chlorophyll content and water loss. Three-week-old seedlings in soil were treated with 150 mM NaCl or 200 mM mannitol for one week. Photographs were taken, and the chlorophyll content and water loss were measured. The error bars represent the standard deviations of the mean measurements, which were calculated from three independent experiments. * indicates a significant difference (*p* < 0.05) between transgenic lines and WT plants.

**Figure 4 plants-11-02647-f004:**
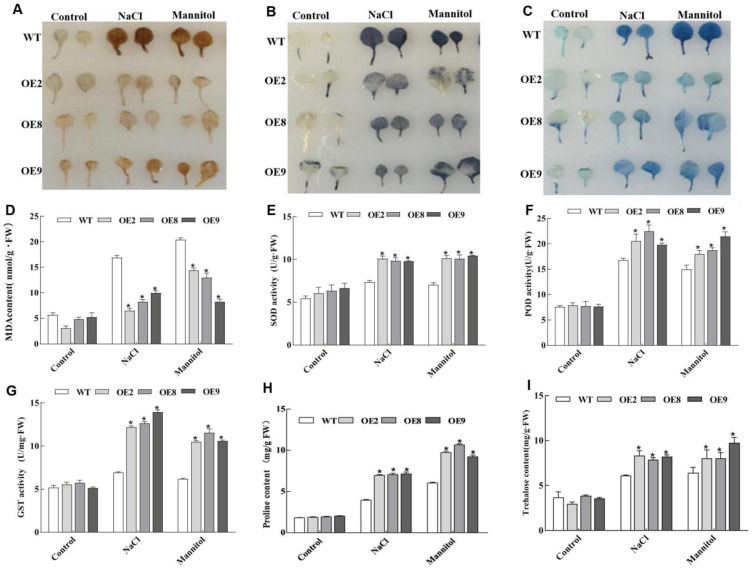
Histochemical staining and physiological analysis of *ThNAC4*-transformed and WT *Arabidopsis* plants. (**A**,**B**) Detection of ROS levels in *ThNAC4*-transformed (Lines 2, 8, and 9) and WT plants. Leaves from transgenic lines and WT plants untreated and treated with 150 mM NaCl or 200 mM mannitol for 2 h were used for histochemical staining by NBT and DAB to reveal the accumulation of O_2_^-^ and H_2_O_2_, respectively. (**C**) Evans blue staining analysis of cell death. Leaves sampled from 4-week-old transgenic and WT seedlings untreated and treated with 150 mM NaCl or 200 mM mannitol for 2 h were used for histochemical staining. (**D**–**I**) Physiological analysis of *ThNAC4*-transformed and WT *Arabidopsis* plants. Four-week-old seedlings of transgenic lines and WT plants were irrigated with 150 mM NaCl or 200 mM mannitol for 2 days to measure MDA contents (**D**); SOD (**E**), POD (**F**), and GST (**G**) activities; and proline (H) and trehalose (**I**) contents. The error bars represent the standard deviations of the mean measurements, which were calculated from three independent experiments. * indicates a significant difference (*p* < 0.05) between transgenic lines and WT plants.

**Figure 5 plants-11-02647-f005:**
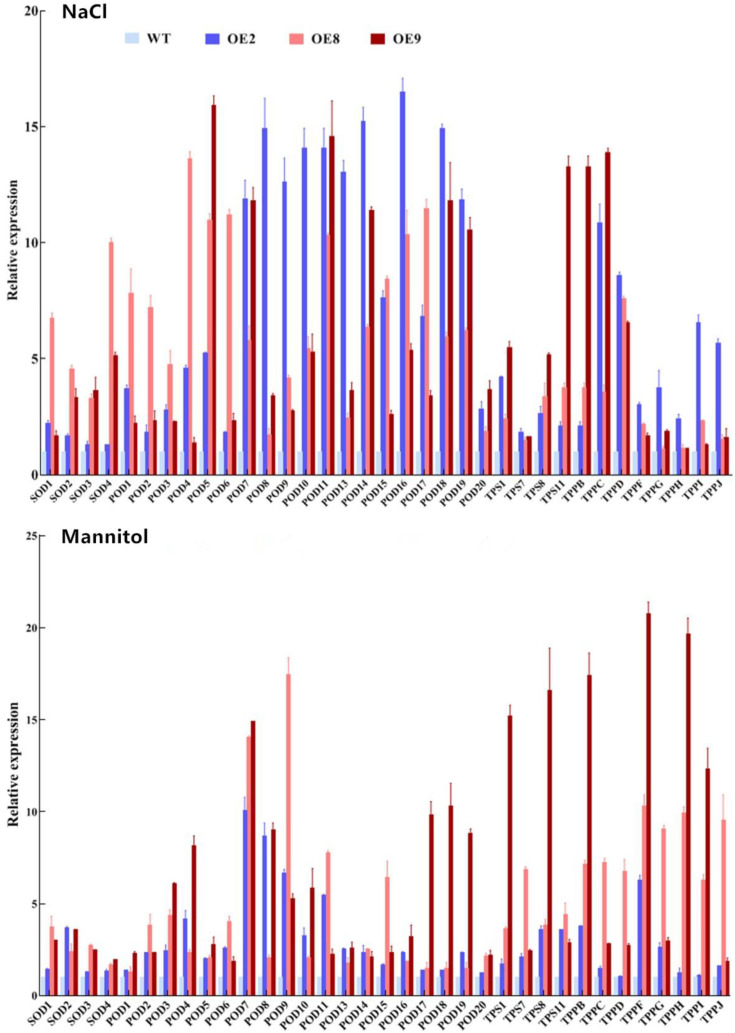
The expression pattern of the *SOD*, *POD*, *TPS* and *TP**P* genes in transgenic and WT *Arabidopsis* plants. Four-week-old seedlings of transgenic lines and WT plants were treated with 150 mM NaCl or 200 mM mannitol for 24 h and were harvested for qRT-PCR analysis. The transcription levels of *SODs, PODs*, *TPSs* and *TP**Ps* in WT plants under the same conditions were set to 1 to calculate their expression in transgenic plants. The error bars were calculated from three replicates.

**Figure 6 plants-11-02647-f006:**
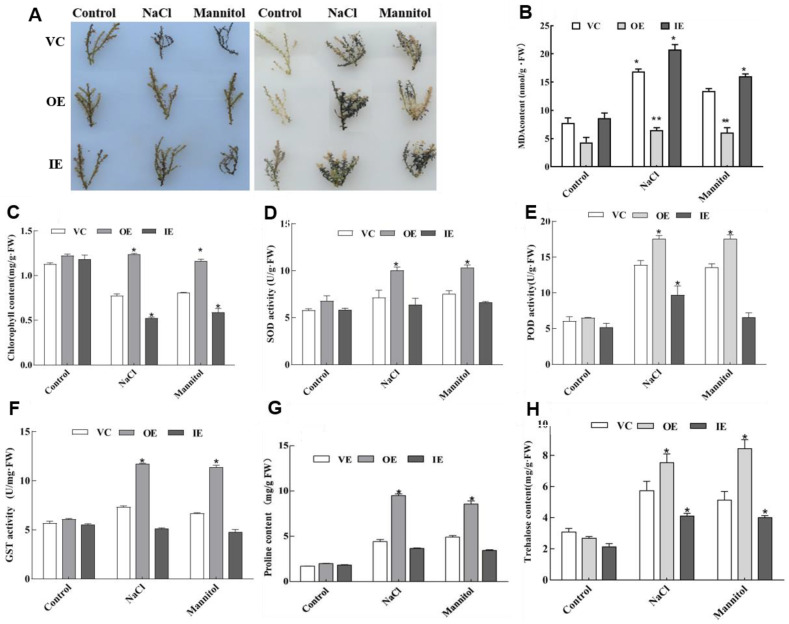
Histochemical staining and physiological analyses in transgenic *T. hispida* plants. (**A**) Detection of ROS levels in transgenic *T. hispida* plants. Histochemical staining by DAB and NBT revealed the accumulation of H_2_O_2_ and O_2_^-^, respectively. (B–H) Physiological analysis of transgenic *T. hispida* plants, including MDA contents (**B**), chlorophyll contents (**C**), SOD activity (**D**), POD activity (**E**), GST activity (**F**), proline contents (**G**) and trehalose contents (**H**). The plants were grown in 1/2 MS medium or 1/2 MS medium containing 150 mM NaCl or 200 mM mannitol for 24 h and used for analysis. VC: the pROKII vector control transformed *T. hispida* plants; OE: overexpression of *ThNAC4* in *T. hispida* plants; and IE: *ThNAC4* RNAi-silenced *T. hispida* plants. The error bars represent the standard deviations of the mean measurements, which were calculated from three independent experiments. *, ** indicates a significant difference (* *p* < 0.05, ** *p* < 0.01) compared to the control plants.

## Data Availability

The data presented in this study are available on request from the corresponding author.

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
