# Peer review of "Tamarix hispida NAC Transcription Factor ThNAC4 Confers Salt and Drought Stress Tolerance to Transgenic Tamarix and Arabidopsis"

_plants, 2022, doi:10.3390/plants11192647_

Round 1

Reviewer 1 Report

       In this manuscript, the authors cloned a NAC gene, ThNAC4, from T.hispida and do the phylogenetic analysis with all NAC proteins from Arabidopsis. They demonstrated that ThNAC4 was located in the nucleus and had transcriptional activation activity, which indicated that ThNAC4 functioned as transcriptional activator. Ectopic overexpression of ThNAC4 in Arabidopsis and T,hispida could improve plants tolerance to salt and drought stress through increase reactive oxygen species scavenging capabilities and osmolytes content.

         However, this study is lack of novelty even though lots of work have been done. Some experiments and analysis are designed irrationally. From conceptualization, methodology to figure pattern, this study is almost the same as Wang et al (2017) with the difference of ThNAC13 instead of thNAC4, while this manuscript discussed nothing about the phylogenetic and functional relationship between the two genes. It is hard to say two different genes have total the same function without redundancy. Obviously, authors didn't mention thNAC13 in this manuscript with the purpose of improving the novelty of ThNAC4.

     The following are some major issues:

1. The authors should provide the transcriptome analysis information or reference to detail how to clone the ThNAC4 full-length cDNA. The accession No. and sequence information about ThNAC4 should be provide.

2.       The authors only studied the evolutionary relationships between ThNAC4 and other 97 NACs from Arabidopsis, which provide little information about NACs evolution and function in T.hispida. I suggest all the NACs from T.hispida should be analyzed, especially ThNAC13.

3.      Line 99-100: ‘we transformedThNAC4-GFP fusion proteins into onion epidermal cells by particle bombardment’. However, in the method (Line 362-363), ‘The two kinds of plasmids were transformed into onion epidermal cells by the Agrobacterium-mediated transformation method’. I just want to ask, which description of method was correct and who performed the experiment?

4.   Figure 3C:  It is obvious that it is blank in the lanes of negative, M1 and M2 on the TDO 3-AT plate panel. Normally, there would be the trace of yeast corps on the TDO 3-AT if the negative control yeast with the same OD as the positive lines was dotted to the TDO plate. I strongly doubt that nothing rather than negative control, M1 and M2 yeast dotted on the TDO plate.

5.   Figure S3. It seemed that ThNAC4 expression in IE line is higher than control plant (designed as 0) under normal growth conditions. I don’t think authors successfully get the RNAi knockdown line. Besides, ThNAC13 expression level should be tested by qRT-PCR.

6.    The detailed method for detecting that ThNAC4 Binds to NACRS and CBNACBS is not provided.

7.    The detailed method for measuring the GST activity is not provided in the method section.

8.    Please provide the detailed method that how to do the Transient Overexpression or Knockdown of ThNAC4 in T. Hispida Plants. It is important to understand the experiments the authors performed to analysis the overexpression or knockdown T. Hispida plant on abiotic stress.

     Minor issues

          Line 354: what is “PIPs”?

Reviewer 2 Report

My comments can be found in the attached PDF.

Round 2

Reviewer 1 Report

In the revised manuscript from Meiheriguli Mijitiet al., the authors made some modifications and re-organized the figures and text to address the concerns raised in the first-round reviews. However, there are still some issues need to be addressed before publication:

1.       Line 482-482: It is not a common method to introduce plasmids into onion epidermal cell by the Agrobacterium mediated transformation. Please provide the resource or reference for this method.

2.       Figure 3C: It can be seen clearly that, on the TDO 3-AT plate, there is only one lane of yeast to be dotted for M1 and M2 samples. Control groups are very important for the experiment. I hope the authors can repeat this experiment and take it seriously. 

Reviewer 2 Report

The Authors have improved the manuscript, although it still lacks a broader perspective.
